# Intensity of Maternal Anxiety and Depressive Symptoms in Pregnancy Is Associated with Infant Emotional Regulation Problems

**DOI:** 10.3390/ijerph192315761

**Published:** 2022-11-26

**Authors:** Alessandra Frigerio, Massimo Molteni

**Affiliations:** Scientific Institute, IRCCS E. Medea, Child Psychopathology Unit, Bosisio Parini, 23842 Lecco, Italy

**Keywords:** antenatal maternal stress, infant crying, negative affect, SES

## Abstract

This study aimed to investigate the effects of the intensity and directionality of antenatal maternal depressive and anxiety symptoms on infant negative affectivity and crying, also taking into account potential confounders. The role of socioeconomic status (SES) as a possible moderating factor of the association between antenatal maternal distress and infant negative outcomes was also explored. More than one hundred women filled in the Edinburgh Postnatal Depression Scale and the State-Trait Anxiety Inventory to assess depressive and anxiety symptoms, respectively, during the third trimester of pregnancy and three months after delivery. Mothers also filled in the Infant Behavior Questionnaire and a parental diary to evaluate negative affectivity and crying, respectively, when their infants were 3 months old. SES was assessed through the Hollingshead classification. The intensity of antenatal maternal symptoms and SES were associated with infant negative affectivity, but not with crying. However, SES moderated the association between the intensity of maternal symptoms and infant crying. The direction of maternal symptoms (anxiety versus depression) was not associated with both infant negative affectivity and crying. Our findings contribute to elucidating the role played by the intensity of maternal stress in pregnancy—alone and in interaction with SES—in determining individual differences in infant emotional regulation, thus emphasizing the importance of timely psychological interventions for pregnant women who experience psychological distress.

## 1. Introduction

Depressive and anxiety symptoms are often co-occurring and represent the most common mental health problems during pregnancy, with prevalence rates ranging from 7 to 22% [1,2,3]. These have possible negative implications not only for maternal health and wellbeing, but also for several areas of child development, appearing from early infancy. Among potential adverse offspring outcomes of antenatal maternal depressive and anxiety symptoms, early negative reactivity represents one of the most crucial in light of its role in predicting later development of psychopathology, e.g., [4,5]. Some studies have shown that antenatal maternal depressive and anxiety symptoms are associated with difficult temperament in the first months of life and, particularly, with infant negative affect [6,7,8]. Negative affect in infancy is usually defined by the presence of emotional distress, irritability, fearfulness, withdrawal in response to novelty and irregularity of biological rhythms. Similarly, there is evidence that excessive infant crying, which is considered one component of negative reactivity, is related to antenatal maternal psychological symptoms [9,10], especially anxiety symptoms [11]. 

However, as far as we know, the effects of maternal depressive and anxiety symptoms—and their co-occurrence—during pregnancy on both infant negative affect and excessive crying have not yet been investigated. In the present study, in order to take into account the comorbidity between anxiety and depression symptoms, we distinguished the total symptom level (i.e., symptom severity) from having a preponderance of anxiety versus depression symptoms (i.e., symptom directionality), in line with previous investigations [12,13] also conducted by our group [14]. 

It is also worth noting that, while direct associations between antenatal maternal depressive/anxiety symptoms and emotional regulation problems in children have been reported, several psychosocial factors, including socioeconomic (SES) disadvantages, might further complicate this picture. There is empirical evidence that SES is associated with maternal depression and anxiety during pregnancy [15,16] and infant emotional regulation problems [17], but whether family SES interacts with antenatal maternal distress in predicting subsequent child emotional regulation problems needs to be clarified. To our knowledge, only Melchior and colleagues [18] have tested the hypothesis that the association between antenatal stress, indexed by maternal depression, and infant temperament could be compounded by low SES (as assessed by having a family income below 1500 EUR/month) in a large community sample. The authors found that 1-year-old children coming from low-income families and who were also exposed to maternal depression in pregnancy tended to be particularly emotional, showing negativity and intense emotional reactions as assessed through a maternal report. 

The current study aimed to investigate the impact of the intensity and directionality (depression versus anxiety) of maternal psychological symptoms during pregnancy on infant negative affectivity and crying, and to explore the potential moderating role of SES. Importantly, the weight of postnatal maternal symptomatology and other potential confounders related to sociodemographic, obstetric and health-related variables was also taken into account in the analyses. No a priori predictions were formulated due to the lack of studies adopting this approach with regard to this topic. 

## 2. Materials and Methods

### 2.1. Sample and Procedure 

The sample of this study is part of an ongoing longitudinal project whose main objective is to evaluate the effects of maternal stress during pregnancy on child development from infancy to childhood in the general population, e.g., [14,19,20]. Only the instruments employed in this study are described here. 

A total of 110 mothers, mostly of Italian descent, were recruited during the third trimester of pregnancy (mean = 34.76; SD = 1.12 weeks) at three hospitals located in the provinces of Como and Lecco. Maternal anxiety and depressive symptoms were assessed through the State-Trait Anxiety Questionnaire (STAI-S) [21] and the Edinburgh Postnatal Depression Scale (EPDS) [22], respectively. Mothers were included in the study if they did not have any other diseases, including psychiatric disorders (with the only exceptions being anxiety and depression) as assessed through the Structured Clinical Interview for DSM-IV-TR Axis I Disorders [23], or were taking any chronic medication. 

Two mothers were excluded from the study at a later point, due to intrauterine death and newborn health problems. At three months after delivery, maternal symptoms were reassessed through the same questionnaires and data on infant crying and temperament were collected, respectively, through the Baby’s Day Diary [24] and the Infant Behavior Questionnaire [25]. At this postnatal phase, 107 mothers filled in the STAI-X, 106 the EPDS and the diary, and 108 the IBQ. Data on socioeconomic status were collected through the Hollingshead scale [26] for parental occupation and were available for 101 participants. 

Infants (51.8% males) were born full-term, except for two late preterm in good health, and mostly by vaginal delivery (82.7%). 

Sociodemographic, obstetric, and health-related variables of the sample are displayed in Table 1. 

The study protocol was approved by the Ethics Committee of Scientific Institute Eugenio Medea and informed written consent to take part in the study was signed by both parents. 

### 2.2. Measures

Edinburgh Postnatal Depression Scale (EPDS) [22]. The EPDS is a 10-item self-report instrument that measures both the presence and severity of depressive symptoms experienced during the past week on a 4-point Likert scale. It has also been validated in Italian [27].

State-Trait Anxiety Inventory–State subscale (STAI-S) [21]. The STAI-S is composed of 20 items that assess current symptoms of anxiety on a 4-point Likert scale. It has also been validated in Italian [28]. 

Infant Behavior Questionnaire (IBQ) [25]. The IBQ (very short form) is a questionnaire, filled in by the caregiver, to assess the temperament of infants between the age of 3 and 12 months. As we were interested in investigating infants’ emotional regulation, the Negative Affectivity subscale of the IBQ was employed. It is composed of 12 items, rated on a 7-point scale, encompassing traits of sadness, distress to limitations, fear, and negatively, falling reactivity.

Baby’s Day Diary [24]. The Italian translation of the Baby’s Day Diary was employed to record infant behaviors (i.e., awake and alert, awake and fussing, awake and crying, awake and inconsolable crying, feeding and sleeping) over a period of 24 h. Specifically, parents are asked to record the onset and end time of consecutive periods of infant behaviors during morning (6 a.m. to 12 p.m.), afternoon (12 p.m. to 6 p.m.), evening (6 p.m. to 12 a.m.) and night (12 a.m. to 6 a.m.). The smallest unit of time that can be recorded is 5 minutes. In line with Bolten et al. [9], the total minutes of fussing, crying and inconsolable crying were summed to create a total score. 

Hollingshead scale [26]. This scale allows SES to be coded on the basis of parental occupation. Specifically, each job is rated on a scale from 10 (e.g., farm laborers, dish washers) to 90 (e.g., engineers, economists), while a score of 0 is assigned when information cannot be scored (e.g., housewives, retired). If both parents are employed, the highest of the two scores is given. 

### 2.3. Statistical Analyses

The average of the STAI-S and EPDS standardized scores was calculated to rate the Intensity of maternal symptoms, while the half difference of the STAI-S and EPDS standardized scores was calculated to rate the Directionality of maternal symptoms, with a positive score showing a predominance of anxiety symptoms [12].

Preliminary Pearson bivariate correlations and univariate analysis of variance (ANOVA) were conducted to identify possible confounding variables, including sociodemographic factors (i.e., maternal and paternal age, maternal and paternal education), obstetric factors (i.e., parity, delivery, gestational age, birth weight) and health-related factors (i.e., smoking and alcohol in pregnancy), that could be associated with infant crying and negative affectivity. All analyses were also adjusted for infant gender [29] and postnatal maternal symptoms. 

Four hierarchical multivariate regression analyses were conducted to analyze the independent and interactive effects of maternal psychological symptoms (intensity and direction) in pregnancy and SES on infant outcomes (i.e., negative affectivity and infant crying scores), also taking into account the effects of covariates. Specifically, confounding variables were entered in the first step, intensity/direction of antenatal maternal psychological symptoms and SES in the second step, and the interaction between the two main predictors (i.e., intensity/direction of antenatal maternal symptoms X SES) in the third step of the analysis. The main independent variables (i.e., intensity of antenatal maternal symptoms, direction of antenatal maternal symptoms and SES) were centered around the mean. 

The 27th version of IBM SPSS Statistics (IBM, Armonk, NY, USA) [30] was used to perform all statistical analyses. 

## 3. Results

### 3.1. Preliminary Analyses 

Pearson bivariate correlations between potential confounding variables (i.e., maternal and paternal age, gestational age, birth weight) and infant outcomes (i.e., negative affectivity and infant crying) were all not significant, as shown in Table 2. 

Furthermore, ANOVAs showed no significant effects on negative affectivity related to maternal (F = 0.04; *p* = 0.84) and paternal education (F = 0.00; *p* = 0.99), type of delivery (F = 0.10; *p* = 0.75), smoking (F = 3.15; *p* = 0.08) and alcohol (F = 0.34; *p* = 0.56), with the only exception being parity (F = 10.74; *p* = 0.001), revealing lower scores in first born infants. No significant effects on infant crying emerged related to maternal (F = 2.22; *p* = 0.14) and paternal education (F = 0.83; *p* = 0.36), type of delivery (F = 1.76; *p* = 0.19), parity (F = 0.69; *p* = 0.41), smoking (F = 0.86; *p* = 0.36) and alcohol (F = 0.002; *p* = 0.97).

As a consequence, only gender, postnatal maternal symptoms and parity (for the negative affectivity scale) were considered as covariates in the main analyses.

### 3.2. Associations between Antenatal Maternal Symptoms and Infant Emotional Regulation Problems

As shown in Table 3, intensity of antenatal maternal symptoms (β = 0.22, *p* = 0.04) and SES (β = 0.24, *p* = 0.01) were significantly associated with increased Negative Affectivity, while adjusting for covariates. On the contrary, no main effects of both predictors on infant crying were found, while a significant effect of the interaction between intensity of antenatal maternal symptoms and SES emerged (β = −0.32, *p* = 0.005), while adjusting for confounders. For illustrative purposes, SES scores were grouped into “below median” and “above median”. As shown in Figure 1, at higher intensity of antenatal maternal symptoms, infants belonging to lower SES showed more excessive crying compared to infants belonging to higher SES. No significant interaction effect between the intensity of antenatal maternal symptoms and SES on Negative Affectivity was found. 

Lastly, the directionality of antenatal maternal symptoms, SES and their interaction were not significantly associated with increased negative affectivity and crying, while controlling for covariates (see Appendix A).

## 4. Discussion

This longitudinal study has examined the link between the intensity and direction (anxiety versus depression) of antenatal maternal psychological symptoms and negative affectivity as well as intensive crying in 3-month-old infants, also exploring the role of SES in moderating these associations. 

We found that the intensity of maternal anxiety and depressive symptoms in pregnancy was positively associated with negative affectivity, independently of infant gender, parity and especially the intensity of postnatal maternal symptoms. There are both biological and psychological processes and mechanisms that may explain this result. In the first case, along with the “weight” of shared genetic susceptibility and epigenetic factors in influencing the relationship between antenatal maternal stress and infant early self-regulation capacity, our finding provides further support to the fetal programming hypothesis [31]. Indeed, it can be supposed that maternal psychological distress experienced in pregnancy can alter the environment in utero and, consequently, impact fetal development, with significant costs for the physical and mental health of the offspring [32,33]. Specifically, although in utero biological mechanisms underlying the association between antenatal maternal distress and infant temperament traits, such as negative affectivity, are mostly unknown, there is preliminary evidence that an altered functioning of the placenta [34], the fetal hypothalamic–pituitary–adrenal axis [35] or alteration of brain morphology [36] might contribute to explaining this association. In the second case, psychological mechanisms mainly related to the low quality of postnatal maternal care in mothers who experience antenatal maternal psychological distress [37] might also explain our finding. Indeed, some studies showed that low maternal sensitivity may amplify the effect of prenatal stress, whereas sensitive caretaking may buffer its impact on child behavioral and emotional development [16,38,39]. Additionally, a low level of social support perceived by mothers during the perinatal period might play a significant role in the association between antenatal maternal depressive/anxiety symptoms and infant negative affectivity. Specifically, it can be hypothesized that distressed mothers who cannot count on someone to confide in and receive parenting advice from might be more likely to engage in negative parenting practices, compared to distressed mothers with high levels of perceived social support, with a subsequent impact on infant behaviors.

Furthermore, we found that family SES was positively associated with infant negative affectivity, also taking into account confounding factors. In other words, mothers coming from higher SES backgrounds, as determined by parental occupations, rated their infants as less able to regulate their emotional states. Although counterintuitive, this finding may show the tendency of high SES parents to be more knowledgeable about child development and, therefore, aware of their difficulties compared to low SES parents, in line with previous research [40]. However, it is important to acknowledge that we assessed family SES only on the basis of parental occupation, without taking into account how other unmeasured SES factors (e.g., parental education, income) might have affected our findings. 

Interestingly, we also found that the intensity of antenatal maternal psychological symptoms and SES were not associated with infant excessive crying, while their interaction was. Specifically, we found that family SES moderated the association between the intensity of maternal depressive and anxiety symptoms during pregnancy and infant crying, which means that infants of mothers who experienced higher depressive and anxiety symptoms showed more excessive crying if they belonged to a lower SES background compared to infants belonging to a higher SES background. Therefore, consistent with the “cumulative stress” hypothesis [41], it might be assumed that infants are more likely to suffer from excessive crying as adversity (i.e., antenatal maternal distress and lower SES levels) accumulates. On the other hand, it is worth noting the role played by belonging to a higher SES background in buffering the impact of the intensity of antenatal maternal psychological distress on infant excessive crying. 

Lastly, we found no effect of the direction of antenatal maternal psychological symptoms, alone and in interaction with SES, on both infant negative affectivity and excessive crying. Therefore, it is likely that depressive and anxiety symptoms in pregnancy might be a broad risk phenotype for child development, in line with previous findings of our group [14]. However, further studies with larger low-risk as well as clinical samples are needed to replicate these preliminary data. 

This study has several methodological strengths including its longitudinal design, the use of validated measures of mother and infant psychological symptoms, and control for a wide range of potential confounders. However, several limitations need to be taken into account. First, we focused on maternal psychological symptoms in a low-risk sample, thus limiting the generalizability of our results to the clinical population. Second, data related to maternal depression and anxiety symptoms as well as to infant negative affectivity and crying are based on maternal self-report or observation, so possible rater-related bias cannot be ruled out. Third, only a single index of SES was used, while employing different measures of SES is recommended [42]. However, it is important to note that some studies reported that, when SES is derived from a single factor, parental occupation is a more useful measure than education [43]. Last, the small contribution of the variables examined and the correlational nature of our results mean our findings should be interpreted cautiously. 

## 5. Conclusions

This study shows the impact of the intensity of maternal psychological symptoms experienced in pregnancy and SES—alone and in interaction—on infant emotional regulation problems. As a consequence, our findings have significant implications for preventive interventions, considering that both negative affectivity and intensive crying in the first months of life have important links to later development of psychopathology. 

## Figures and Tables

**Figure 1 ijerph-19-15761-f001:**
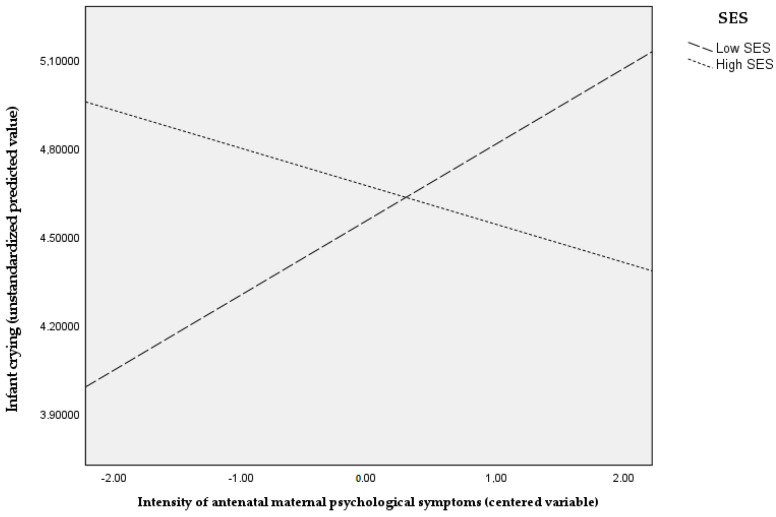
Graphical representation of the interaction effect (antenatal maternal psychological symptoms × SES) on infant crying.

**Table 1 ijerph-19-15761-t001:** Sociodemographic, obstetric, and health-related and study variables.

	N	%	
Sociodemographic, obstetric and health-related variables (N = 110)			
Child gender (male)	57	51.8	
First born (yes)	99	90.0	
Marital status (married)	69	62.7	
Maternal education (>10 years)	99	90.0	
Paternal education (>10 years)	88	80.0	
Delivery (vaginal)	91	82.7	
Smoking (yes) *	13	12.1	
Alcohol (yes)	25	22.7	
	Mean	SD	Range
Child age (days)	88.61	11.59	68–144
Birth weight (grams)	3283.62	438.104	2170–4240
Maternal age (years)	33.04	3.75	27–45
Paternal age (years)	35.96	4.90	27–56
Gestational age (weeks)	39.44	1.25	35–42
Maternal symptoms and SES			
STAI-S pregnancy	34.66	7.92	20–57
EPDS pregnancy	5.01	3.86	0–19
STAI-S 3 months	32.60	8.26	20–66
EPDS 3 months	4.66	3.62	0–22
SES	64.36	18.05	20–90
Infant outcomes			
Infant crying (minutes)	116.94	77.39	0–335
Negative Affectivity	3.51	0.94	1.58–5.75

Abbreviations: STAI-S: State Trait Anxiety Inventory–State subscale; EPDS: Edinburg Postnatal Depression Scale; SES: socioeconomic status; * Percentages for smoking do not add up to 100% due to missing values.

**Table 2 ijerph-19-15761-t002:** Pearson correlations among sociodemographic variables, maternal prenatal and postnatal symptoms and infant outcomes.

	1	2	3	4	5	6	7	8	9	10	11
Birth weight	-										
2. Maternal age	−0.07	-									
3. Paternal age	−0.13	**0.55 *****	-								
4. Gestational age	**0.42 *****	0.16	0.03	-							
5. SES	−0.17	0.01	−0.01	0.01	-						
6. Intensity of prenatal symptoms	0.02	0.03	**−0.23 ***	−0.01	−0.16	-					
7. Directionality of prenatal symptoms	0.04	0.04	0.03	0.02	−0.04	−0.02	-				
8. Intensity of postnatal symptoms	0.17	−0.13	−0.12	0.09	−0.12	**0.40 *****	0.03	-			
9. Directionality of postnatal symptoms	−0.06	0.02	0.08	−0.04	−0.07	−0.04	**0.23 ***	−0.05	-		
10. Infant crying	−0.15	−0.02	0.05	−0.04	0.06	0.06	−0.19	0.15	−0.09	-	
11. Negative affectivity	0.06	0.13	0.11	−0.01	**0.21 ***	**0.29 ****	0.06	**0.23 ***	0.05	**0.26 ****	-

Values in bold indicate statistically significant results; * *p* < 0.05; ** *p* < 0.01, *** *p* < 0.001.

**Table 3 ijerph-19-15761-t003:** Intensity of antenatal maternal symptoms and SES as predictors of infant crying and negative affectivity.

	Negative Affectivity		Infant Crying	
	*β*	*p*	*β*	*p*
Step 1:				
Gender	0.12	0.23	−0.09	0.43
First born	**−0.32**	**0.002**		
Intensity of postnatal maternal symptoms	**0.21**	**0.03**	0.17	0.12
	*R*^2^ = 0.15		*R*^2^ = 0.04	
Step 2:				
Intensity of antenatal maternal symptoms	**0.22**	**0.03**	0.01	0.95
SES	**0.24**	**0.01**	0.07	0.50
	Δ*R*^2^ = 0.09		Δ*R*^2^ = 0.01	
Step 3:				
Intensity of antenatal maternal symptoms	**0.22**	**0.04**	−0.03	0.79
SES	**0.24**	**0.01**	0.11	0.29
Intensity of antenatal maternal symptoms × SES	−0.06	0.57	**−0.32**	**0.005**
	Δ*R*^2^ = 0.003		Δ*R*^2^ = 0.09	

Bold values indicate significant *p* < 0.05 results. SES: socioeconomic status.

## Data Availability

The dataset generated for this study is available upon request to the corresponding author.

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
