# Peer review of "Intensity of Maternal Anxiety and Depressive Symptoms in Pregnancy Is Associated with Infant Emotional Regulation Problems"

_ijerph, 2022, doi:10.3390/ijerph192315761_

Round 1

Reviewer 1 Report

The study investigates the possible relationship between antenatal mothers' depressive and anxiety symptoms during the third trimester of pregnancy and the later infant emotional regulation problems. I found the study original and interesting, grounded on solid background and characterized by a strong study design. For these reasons, I suggest the publication of the study in the IJERPH.

I have just a minor suggestion/ consideration. the authors considered the moderator effect of SES on the relationship analysed. As the authors highlight in the discussion, a limitation of the study is represented by the use of a single measure in determining the SES. Along with this point, I think it would be of interest considering also the social support perceived by mothers as moderator of the relationships investigated. If the authors have data about this point, it suggest to add it. if this data is not available, I suggest to insert a consideration about it within the discussion.

Author Response

The study investigates the possible relationship between antenatal mothers' depressive and anxiety symptoms during the third trimester of pregnancy and the later infant emotional regulation problems. I found the study original and interesting, grounded on solid background and characterized by a strong study design. For these reasons, I suggest the publication of the study in the IJERPH.

We thank the reviewer for his/her positive comments to our manuscript.

I have just a minor suggestion/ consideration. the authors considered the moderator effect of SES on the relationship analysed. As the authors highlight in the discussion, a limitation of the study is represented by the use of a single measure in determining the SES. Along with this point, I think it would be of interest considering also the social support perceived by mothers as moderator of the relationships investigated. If the authors have data about this point, it suggest to add it. if this data is not available, I suggest to insert a consideration about it within the discussion.

We thank the reviewer for this suggestion. We fully agree that social support might be a potential moderating variable of the association between maternal psychological distress in pregnancy and infant negative outcomes. However, this kind of investigation was beyond the scope of the current study. Consequently, as suggested by the reviewer, we reported the possible moderating role of perceived social support in the Discussion (page 7) as follows:

Additionally, a low level of social support perceived by mothers during the perinatal period might play a significant role in the association between antenatal maternal depressive/anxiety symptoms and infant negative affectivity. Specifically, it can be hypothesized that distressed mothers who cannot count on someone to confide in and receive parenting advice from might be more likely to engage in negative parenting practices, compared to distressed mothers with high levels of perceived social support, with a subsequent impact on infant behaviors.”

Reviewer 2 Report

I would like to congratulate the authors for the excellent manuscript. The article is well written, with an excellent theoretical foundation. The methodology is detailed and the results are well described. The discussion is grounded in and related to the results of the study.

I have some considerations to point out about the abstract:

The first sentence is too long, I suggest reviewing it (This study aimed to investigate the effects of intensity and directionality of antenatal maternal depressive and anxiety symptoms on infant negative affect and crying as well as the role of socioeconomic status as a possible moderating factor, taking also into account potential confounders.) ;

An explanation of how the data analysis was performed would be interesting for a better understanding of the readers.

I also suggest adding some numerical data to illustrate the results.

Author Response

I would like to congratulate the authors for the excellent manuscript. The article is well written, with an excellent theoretical foundation. The methodology is detailed and the results are well described. The discussion is grounded in and related to the results of the study.

We are very grateful to the reviewer for his/her positive evaluation of our work.

I have some considerations to point out about the abstract:

The first sentence is too long, I suggest reviewing it (This study aimed to investigate the effects of intensity and directionality of antenatal maternal depressive and anxiety symptoms on infant negative affect and crying as well as the role of socioeconomic status as a possible moderating factor, taking also into account potential confounders.) ;

We have now shortened the first sentence of the abstract as follows: This study aimed to investigate the effects of the intensity and directionality of antenatal maternal depressive and anxiety symptoms on infant negative affectivity and crying, also taking into account potential confounders. The role of socioeconomic status (SES) as a possible moderating factor of the association between antenatal maternal distress and infant negative outcomes was also explored”.

An explanation of how the data analysis was performed would be interesting for a better understanding of the readers.

In the statistical analyses paragraph (page 4), we have now better specified how the main analyses (i.e. hierarchical regression analyses) were performed as follows:  

Four hierarchical multivariate regression analyses were conducted to analyze the independent and interactive effects of maternal psychological symptoms (intensity and direction) in pregnancy and SES on infant outcomes (i.e., negative affectivity and infant crying scores), also taking into account the effects of covariates. Specifically, confounding variables were entered in the first step, intensity/direction of antenatal maternal psychological symptoms and SES in the second step, and the interaction between the two main predictors (i.e., intensity/direction of antenatal maternal symptoms X SES) in the third step of the analysis. The main independent variables (i.e., intensity of antenatal maternal symptoms, direction of antenatal maternal symptoms and SES) were centered around the mean”.

I also suggest adding some numerical data to illustrate the results.

We thank the reviewer for pointing this out. With regard to Preliminary Analyses, we have now added the F and p values related to ANOVA’s results as follows (page 5):

Furthermore, ANOVAs showed no significant effects on negative affectivity related to maternal (F = 0.04; p = 0.84) and paternal education (F = 0.00; p = 0.99), type of delivery (F = 0.10; p = 0.75), smoking (F = 3.15; p = 0.08) and alcohol (F = 0.34; p = 0.56), with the only exception being parity (F = 10.74; p = 0.001), revealing lower scores in first born infants. No significant effects on infant crying emerged related to maternal (F = 2.22; p = 0.14) and paternal education (F = 0.83; p = 0.36), type of delivery (F = 1.76; p = 0.19), parity (F = 0.69; p = 0.41), smoking (F = 0.86; p = 0.36) and alcohol (F = 0.002; p = 0.97)”

while, with regard to main analyses, all statistical values had already been reported in the text or in the tables.

Reviewer 3 Report

First of all, I would like to congratulate authors for considering this important aspect of child’s development. Early childhood is sometimes forgotten, and all developmental processes occurred at this period are important for the future development.

Secondly, I would like to comment that at certain points, I found difficult to understand the text due to linguistic mistakes. I am sure that I would appreciate more the manuscript if this would be better written. Therefore, I would recommend improving this aspect.

Finally, I would like to comment an aspect that I have found difficult to defend. I do not understand the rationale to categorize parental occupation into discrete numbers. I can understand the use of the Hollingshead scale for economical research, by categorizing occupations in the basis of the income, for example, or the technical training. But I find difficult to apply this categorization for the purpose of this study. As an example, a businesswoman can be very stressed due to her occupation, and this could influence her pregnancy and her baby’s first months; but probably she would be rated with a 90 in this scale. Of course, if she would be aware of her stress and its implications for the baby, she would find other resources to cope with the situation. My concern is about why not a certain type of occupation, as well as certain personality traits or attachment styles would be influencing, instead of an occupational range per se.

This has implications for the introduction section, as well as for discussion and conclusions. I would appreciate a better rationale to explain why authors have considered this scale, instead of categorizing parental occupation based in another criterion, or considering personality traits or attachment style.

Author Response

First of all, I would like to congratulate authors for considering this important aspect of child’s development. Early childhood is sometimes forgotten, and all developmental processes occurred at this period are important for the future development.

We thank the reviewer for noticing and appreciating our focus on early predictors of child emotional and behavioral problems     

Secondly, I would like to comment that at certain points, I found difficult to understand the text due to linguistic mistakes. I am sure that I would appreciate more the manuscript if this would be better written. Therefore, I would recommend improving this aspect.

English has been revised through the editing service of the publisher.

Finally, I would like to comment an aspect that I have found difficult to defend. I do not understand the rationale to categorize parental occupation into discrete numbers.  I can understand the use of the Hollingshead scale for economical research, by categorizing occupations in the basis of the income, for example, or the technical training. But I find difficult to apply this categorization for the purpose of this study. As an example, a businesswoman can be very stressed due to her occupation, and this could influence her pregnancy and her baby’s first months; but probably she would be rated with a 90 in this scale. Of course, if she would be aware of her stress and its implications for the baby, she would find other resources to cope with the situation. My concern is about why not a certain type of occupation, as well as certain personality traits or attachment styles would be influencing, instead of an occupational range per se.

This has implications for the introduction section, as well as for discussion and conclusions. I would appreciate a better rationale to explain why authors have considered this scale, instead of categorizing parental occupation based in another criterion, or considering personality traits or attachment style.

We thank the reviewer for raising this issue, which allows us to better support the rationale of our study and the reasons for using the Hollingshead scale.

First, we would like to clarify that our choice to investigate the role played by SES as a potential moderating variable in the association between antenatal maternal psychological distress and infant negative affectivity/excessive crying is supported by the literature on these topics. Indeed, if we want to investigate the potential influence of maternal distress on infant negative outcomes, it is crucial to consider the broader context, including family SES, in which the mother-child relationship is embedded. Several studies have shown that low SES is associated with maternal depression/anxiety during pregnancy (e.g. Goyal et al., 2010; Cena et al., 2020) and infant emotional regulation (e.g. Gago-Galvagno et al., 2022). However, to the best of our knowledge, only one study (Melchior et al., 2012) has moved beyond exploring direct associations among these variables to testing whether family SES may interact with antenatal maternal distress in predicting subsequent child emotional regulation problems.

We have now better specified this issue in the Introduction (page 2) as follows:

There is empirical evidence that SES is associated with maternal depression and anxiety during pregnancy [15, 16] and infant emotional regulation problems [17], but whether family SES interacts with antenatal maternal distress in predicting subsequent child emotional regulation problems needs to be clarified”.

  1. Goyal D., Gay C., Lee K.A. How much does low socioeconomic status increase the risk of prenatal and postpartum depressive symptoms in first-time mothers? Women's Health Issues. 2010;20:96–104.
  2. Cena L, Mirabella F, Palumbo G, Gigantesco A, Trainini A, Stefana A. Prevalence of maternal antenatal anxiety and its association with demographic and socioeconomic factors: A multicentre study in Italy. Eur Psychiatry. 2020;63(1):e84.
  3. Gago-Galvagno LG, Miller SE, De Grandis C, Elgier AM, Mustaca AE, Azzollini SC. The still-face paradigm in Latin American mother-child dyads at 2 and 3 years: Effects of socioeconomic status and temperament. J Exp Child Psychol. 2022 May; 217:105357.

Moreover, in the Discussion, we reported other possible biological and psychological factors that can explain the association between antenatal maternal distress and infant negative outcomes and, as requested by Reviewer 1, we have now included the social support perceived by mothers during perinatal period as a possible underlying factor. 

“Additionally, a low level of social support perceived by mothers during the perinatal period might play a significant role in the association between antenatal maternal depressive/anxiety symptoms and infant negative affectivity. Specifically, it can be hypothesized that distressed mothers who cannot count on someone to confide in and receive parenting advice from might be more likely to engage in negative parenting practices, compared to distressed mothers with high levels of perceived social support, with a subsequent impact on infant behaviors”.

Second, we decided to use the Hollingshead classification as it represents one of the most frequently used measures of SES in the social and behavioral sciences (Edwards-Hewitt & Gray, 1995), even within the context of research on parenting behavior (Callahan and Eyberg, 2010) and child psychopathology (Peverill et al. 2022).    

Edwards-Hewitt, T., & Gray, J. J. (1995). Comparison of measures of socioeconomic status between ethnic groups. Psychological Reports, 77, 699-702.

Callahan, C. L., & Eyberg, S. M. (2010). Relations between parenting behavior and SES in a clinical sample: Validity of SES measures. Child & Family Behavior Therapy, 32(2), 125–138. 

Peverill, M.; Dirks, M.A.; Narvaja, T.; Herts, K.L.; Comer, J.S.; McLaughlin, K.A. Socioeconomic status and child psychopathology in the United States: A meta-analysis of population-based studies. Clin. Psychol. Rev. 2021, 83:101933; 

Moreover, an interesting work of Cirino and colleagues (2002) has also investigated “the usefulness of a more simplified approach to SES, using either the unweighted education and or occupation of Hollingshead (1975)”. The results of their study “suggested that when using a more simplified model to derive SES, occupation is a more useful single factor than education. This result is similar to that reported by Gottfried (1985), who noted that occupation from the Hollingshead scale was interchangeable with the other SES measures he examined. Indeed, derivation of SES based on an individual’s occupational category alone, for some purposes, may provide scores that are comparable to those obtained through other more involved or time-consuming approaches”.

  1. Cirino, P.T., Chin, C.E., Sevcik, R.A., Wolf, M., Lovett, M. & Morris, R.D. (2002). Measuring socioeconomic status: Reliability and preliminary validity of different approaches. Assessment, 9(2), 145-155.

Gottfried, A. W. (1985). Measures of socioeconomic status in child development research: Data and recommendations. Merrill-Palmer Quarterly, 31(1), 85-92.

Thus, we have now reported in the Discussion this clarification as follows:

Third, only a single index of SES was used, while employing different measures of SES is recommended [42]. However, it is important to note that some studies reported that, when SES is derived from a single factor, parental occupation is a more useful measure than education [43].

Last, we wish to specify that in the hierarchical multivariate regression analyses, SES was used as a continuous variable (from 0 to 90). Only for illustrative purposes, SES scores were grouped into “below vs above median” in order to better show the effect of the interaction between the intensity of antenatal maternal psychological symptoms and SES on infant crying.